# Effect of Vitamin E Supplement on Bone Turnover Markers in Postmenopausal Osteopenic Women: A Double-Blind, Randomized, Placebo-Controlled Trial

**DOI:** 10.3390/nu13124226

**Published:** 2021-11-25

**Authors:** Sakda Arj-Ong Vallibhakara, Katanyuta Nakpalat, Areepan Sophonsritsuk, Chananya Tantitham, Orawin Vallibhakara

**Affiliations:** 1Faculty of Medicine, Bangkokthonburi University, Bangkok 10170, Thailand; dr.sakda@gmail.com; 2Child Safety Promotion and Injury Prevention Research Center, Faculty of Medicine, Ramathibodi Hospital, Mahidol University, Bangkok 10400, Thailand; 3Woman Health Centre, Chulabhorn Hospital, HRH Princess Chulabhorn College of Medical Science, Chulabhorn Royal Academy, Bangkok 10210, Thailand; Katanyuta_ne@hotmail.com; 4Reproductive, Endocrinology & Infertility Unit, Department of Obstetrics and Gynaecology, Faculty of Medicine Ramathibodi Hospital, Mahidol University, Bangkok 10400, Thailand; areepan.sop@mahidol.ac.th (A.S.); ploy.dango@gmail.com (C.T.)

**Keywords:** vitamin E, bone turnover, osteopenia, postmenopausal women, tocopherol, bone marker turnover, bone health

## Abstract

Vitamin E is a strong anti-oxidative stress agent that affects the bone remodeling process. This study evaluates the effect of mixed-tocopherol supplements on bone remodeling in postmenopausal osteopenic women. A double-blinded, randomized, placebo-controlled trial study was designed to measure the effect of mixed-tocopherol on the bone turnover marker after 12 weeks of supplementation. All 52 osteopenic postmenopausal women were enrolled and allocated into two groups. The intervention group received mixed-tocopherol 400 IU/day, while the control group received placebo tablets. Fifty-two participants completed 12 weeks of follow-up. Under an intention-to-treat analysis, vitamin E produced a significant difference in the mean bone resorption marker (serum C-terminal telopeptide of type I collagen (CTX)) compared with the placebo group (−0.003 ± 0.09 and 0.121 ± 0.15, respectively (*p* < 0.001)). In the placebo group, the CTX had increased by 35.3% at 12 weeks of supplementation versus baseline (*p* < 0.001), while, in the vitamin E group, there was no significant change of bone resorption marker (*p* < 0.898). In conclusion, vitamin E (mixed-tocopherol) supplementation in postmenopausal osteopenic women may have a preventive effect on bone loss through anti-resorptive activity.

## 1. Introduction

Bone is a dynamic tissue continuously remodeling itself throughout life for bone homeostasis [1]. The remodeling process is based on the balanced activities of bone resorption and formation. This process requires communication between different bone cells, namely, cells of the osteoblast lineage (osteoblasts and osteocytes) and bone-resorbing cells (osteoclasts), which are organized in specialized units called bone multicellular units (BMU). The bone remodeling controlled by various local and systemic factors in different pathways, such as calcitonin, parathyroid hormone, and estrogen pathways, is the major hormonal regulator of osteoclastic bone resorption. In addition, cytokines, such as Interleukin-1 (IL-1), Interleukin-6 (IL-6), prostaglandin E2 (PGE2), and tumor necrotic factor α (TNF-α), play roles in the regulation of physiological bone remodeling. The major pathway of bone remodeling is controlled by the receptor activator of nuclear factor kappa-B (RANK), receptor activator of nuclear factor kappa-B ligand (RANKL), osteoprotegerin (OPG), canonical Wnt/β-catenin, and oxidative stress signaling pathways [2,3,4,5]. The oxidative stress pathway activates the differentiation of pre-osteoclast into osteoclast and provokes bone resorption. An increasing level of reactive oxygen species (ROS) results in rapid bone loss, especially in postmenopausal women [6]. 

A postmenopausal woman means a woman with an absence of menstruation of at least 12 months, resulting from the cessation of ovarian function [7]. The resultant reduction of the female reproductive hormone associated with aging process, such as osteoporosis, atherosclerosis, and cancer. Oxidative stress is the pathophysiological driver of the aging process and a cause of osteoporosis. Oxidative stress aggravates the osteoclastic activity in the bone remodeling process, without effect on the osteoblast. Moreover, bone resorption predominates bone formation, resulting in diminished bone density. As mentioned, these mechanisms/factors are part of the cause of osteopenia and osteoporosis in postmenopausal women and the elderly [8,9]. The same as seen in rheumatoid arthritis patients who have a high level of ROS that causes low bone mass even during reproductive age [10,11]. 

Osteopenia, or low bone mass, as defined by the World Health Organization, is when bone mineral density (BMD) value, reported as a T-score and measured by a dual-energy X-ray absorptiometry (DXA), is between T-score-1 and -2.5 [12]. The prevalence of osteopenia is high worldwide, which an estimated 43.9%, 40.3%, and 59.4% of postmenopausal women in the United States, India, and Thailand, respectively [13,14,15]. The relative risk for fracture is three times greater for osteopenic women compared to women having normal BMD. Moreover, Pasco et al. found that the osteoporotic fractures mostly come from osteopenic women, who represent as high as approximately 50% of the total fracture population [16]. The latest survey study in Thailand reveals morphometric vertebral fracture at 29% among postmenopausal women with osteopenia [17]. Although women with osteoporosis are at the highest risk for fracture, these high-risk individuals comprise less than 20% of all women who fracture. Therefore, strengthening bone quality and fall prevention in postmenopausal women with osteopenia, which is not indicated for pharmacologic treatment, are essential for minimizing morbidity and mortality from fractures. According to the American Association of Clinical Endocrinologists/American College of Endocrinology clinical practice guideline for postmenopausal osteoporosis treatment. There is a strong indication for pharmacologic therapy when patient has: (1) low bone mass and a history of a fragility fracture of hip or spine; (2) osteoporosis, as diagnosed by BMD measurement with a T-score of -2.5 or lower in the spine, femoral neck or distal radius; and (3) osteopenia, as diagnosed by BMD measurement with the fracture risk assessment tool (FRAX^®^) 10-year probability for major osteoporosis fracture being ≥20% or the 10-year probability of hip fracture being ≥3%. The first-line medical management for osteoporotic fracture prevention is bisphosphonate agents, including alendronate, risedronate and zoledronate, and the receptor activator of nuclear factor kappa-Β ligand (RANKL) inhibitors, or denosumab [18]. Those medications inhibit osteoclast’s function, which affects the risk of the bone freezing condition, rare but serious complications include osteonecrosis of the jaw (ONJ), and atypical femoral fracture (AFF). The ONJ is the presence of exposed bone in the mouth that fails to heal after appropriate intervention over a period of 6 or 8 weeks. Incidence has been reported between 0.001–0.06% depending on the dose and duration of anti-resorptive administration [19,20]. In contrast, AFF is a transverse fracture of the femoral shaft (diaphysis), defined by clinical criteria and radiographic appearance. The incidence ranged from 3.0 to 9.8 cases per 100,000 patient per year [21,22]. Drug holiday guidelines recommend discontinuing bisphosphonates after five years of treatment, depending on patient fracture risk to prevent those serious side effects. 

Vitamin E, one of the lipid-soluble vitamins, is a well-known strong anti-oxidative agent or antioxidant. This consists of two isoforms: tocopherol (TF) and tocotrienol (T3). Each isoform divides into four distinct analogs, namely alpha (α), beta (β), gamma (γ), and delta (δ). Tocopherol is the most commonly available form of vitamin E supplement on the market. The major properties of vitamin E are reducing the reactive oxygen species (ROS), and numerous pro-inflammatory cytokines, such as IL-1, IL-6, PGE2, and TNF-α, which are in part responsible for the activation of osteoclastic activity [23,24,25,26,27,28]. From observational studies in postmenopausal women, low serum α-tocopherol and low intake of vitamin E in diet were associated with osteoporosis and hip fracture [29,30]. In the animal model study, Muhammad et al. (2012) found that tocopherol and tocotrienol can decelerate bone resorption activity and decrease bone loss [31,32,33]. 

The study aimed to evaluate the effect of vitamin E supplements in postmenopausal osteopenic women on bone by measuring bone turnover marker changes after a mixed-tocopherol supplement for 12 weeks. The investigation was based on the hypothesis that vitamin E could decrease the differentiation and activity of osteoclast that causes bone loss attenuation.

## 2. Materials and Methods 

### 2.1. Study Design

The study protocol conformed to the principles of the Declaration of Helsinki. It was approved by the Ethical Clearance Committee on Human Rights Related to Research Involving Human Subjects Faculty of Medicine Ramathibodi Hospital, Mahidol University (MURA 2018/02) and Grants. The study protocol was also submitted to the Thai Clinical Trials Registry; TCTR (www.thaiclinicaltrials.org) Clinical trial registration no. TCTR20180419001/18-04-2018. The double-blinded, placebo-controlled randomized trial was undertaken to study the effect of mixed-tocopherol on changes in bone turnover marker (serum C-terminal telopeptide of type I collagen (CTX), a bone resorption marker, and serum N-terminal propeptide of type I procollagen (P1NP), a bone formation marker) after 12 weeks of supplementation. The primary outcome was (1) a change of bone resorption marker (CTX) and (2) bone formation marker (P1NP). The secondary outcome was any adverse events.

### 2.2. Participants

Postmenopausal osteopenic women were enrolled in the study between 1 May 2018 and 31 May 2019 at Menopause clinic, Department of Obstetrics and Gynecology, Faculty of Medicine, Ramathibodi Hospital, Bangkok. The informed and consent of study was obtained from all participants. Postmenopausal women with absent menstruation for at least one year, age over 45-year-old, osteopenia indicated by a bone mineral density measured by Dual-energy X-ray Absorptiometry T-score between −1 and −2.5 at the lumbar spine, total hip or femoral neck, and signed agreements to participate in the study were included. Exclusion criteria were serum 25-hydroxyvitamin D (serum 25(OH)D level) < 20 ng/mL, having a history of metabolic bone disease, having a history of cancer, having an eating disorder or malabsorption, taking a medication that affects bone metabolism, taking an anticoagulant, having taken a vitamin E supplement within 3 months, and unwillingness to accept the randomization. A pilot study was performed to estimate sample size from 16 participants. The means and standard deviation (SD) of C-terminal telopeptide of type I collagen (CTX) from both groups were measured and calculated by N4studies Application (Version 1.4.1). The mean in a treatment group (mixed-tocopherol supplement) was 0.35, and SD was 0.15. The mean in a control group (Placebo) was 0.55, and SD was 0.29. The ratio (control/treatment) = 1.00. Alpha (α) = 0.05, Z (0.975) = 1.959964, and Beta (β) = 0.20, Z (0.800) = 0.841621. Then, the calculated sample size was 21 participants in the treatment and an equal number in the control group. A 20% larger sample was used to accommodate for potential data loss. Finally, there were 26 participants in each group, or a total of 52 participants in this study. 

### 2.3. Randomization, Blinding, and Intervention Protocol

All participants were randomly assigned in a 1:1 ratio into two groups by the computerized block of four randomizations. The participants and investigators were blinded to the group allocation. Only the pharmacist, who prepared these study tablets, knew the allocation. Participants in the study group received 400 IU of mixed-tocopherol one time per day for 12 weeks. Each tablet contained 20% delta-tocopherol, 1% beta tocopherol, 62% gamma-tocopherol, and 10% alpha tocopherol (Nat E^®^, Mega Lifesciences Public Company Limited, Samutprakarn, Thailand). The dose of vitamin E was calculated based on a conversion from rat to human by body surface area [34,35]. Participants in the placebo group received the placebo tablets, which had a similar external appearance to the mixed-tocopherol tablets but contained soybean oil. The participants in both groups received 600 mg of calcium carbonate twice per day (total of 1200 mg per day) and 20,000 IU of vitamin D2 one time per week.

### 2.4. Data Collection and Measurements

At the time of enrollment, baseline characteristics, including age, parity, menopause age, time since menopause, medical history, current medication, prior calcium or vitamin D or vitamin E supplement, smoking history, drinking history, and exercise habits, were gathered. Physical examinations were performed, and the clinical data, including weight, height, body mass index, blood pressure, and any signs of bleeding tendency, were collected. For exploring the secondary cause of bone loss, baseline laboratory analysis of serum 25(OH)D level, creatinine, aspartate aminotransferase (AST), alanine aminotransferase (ALT), parathyroid hormone level (PTH), thyroid-stimulating hormone (TSH), and complete blood count (CBC) were performed. 

The change of bone turnover markers (BTMs) represented the bone remodeling process and was the primary outcome of this study. The BTMs are byproducts of the bone remodeling process. They include formative bone marker; bone-specific alkaline phosphatase (B-ALP), procollagen type 1 N-terminal propeptide (P1NP), osteocalcin (O), and procollagen type 1 C-terminal propeptide (P1CP), and bone resorption marker; hydroxyproline (HYP), pyridinoline, tartrate-resistant acid phosphatase 5b (TRAP5b), deoxypyridinoline (DPD), carboxy-terminal cross-linked telopeptide of type 1 collagen (CTX-1), and amino-terminal cross-linked telopeptide of type 1 collagen (NTX-1). Those markers are used in clinical trials since they rapidly change, compared to BMD, and are easily collected from urine or serum. Currently, BTMs have already proved to be of clinical use as adjuvant tools for fragility fracture risk stratification and monitoring the treatment response and adherence monitoring. The bone remodeling process consists of the bone resorption phase, in which the main action is by osteoclasts catabolizing old bone (lasts for 4–6 weeks) and the subsequent bone formation phase in which the osteoblast secretes molecules that fill in cavities with osteoid, a connective tissue rich in collagen. This entire bone formation process lasts 4–5 months [36]. According to the International Osteoporosis Foundation and the International Federation of Clinical Chemistry and Laboratory Medicine Working Group on Bone Marker Standards, serum C-terminal telopeptide (CTX) and serum N-terminal propeptide of type I procollagen (P1NP) have been recommended as the standard marker of bone resorption and formative bone marker, respectively [37]. Serum C-terminal telopeptide (CTX) is a product of the breakdown of type I collagen, the major bone protein component. After the release of the osteoclast enzyme, bone proteins are digested and released as their fragments, e.g., N-telopeptide of type I collagen (NTX-I) or CTX-I, into serum, plasma, and urine. Serum CTX is a bone-specific marker and decreases during anti-resorptive treatment. In contrast, serum N-terminal propeptide of type I procollagen (P1NP) is derived from post-translational cleavage of type I procollagen during the bone formative phase. Procollagen type I enzymatically cleaves the N-propeptides and C-propeptides and subsequently becomes collagen type I, which is deposited in a quarter stagger array held together by the pyridinium cross-links: deoxypyridinoline and pyridinoline. Osteoblasts express their highest collagen concentration during the proliferative phase, bone alkaline phosphatase, which occurs during matrix maturation, and osteocalcin during mineralization. Serum P1NP is a bone-specific marker that weakens circadian variation and increases during bone formation-stimulating therapy [38]. 

In this study, both bone turnover markers were measured by automated electrochemiluminescence immunoassay (ECLIA) for CTX and PINP, with cobas E602 (Roche Diagnostic, Germany). The intra- and inter assays coefficient of variation were 1.6% (0.004 ng/mL) and 2.2% (0.006 ng/mL), for the serum CTX and 1.7% (0.524 ng/mL) and 2.6% (0.797 ng/mL) for PINP, respectively. The venous blood collection was performed in a fasting sample at 7–10 a.m. to minimize the diurnal variation on the day of enrollment and at the end of the 12th week of follow-up. The serum was collected using standard sampling with EDTA tubes.

Participants received a reminder phone call one week before the appointment. At the 12th week of follow-up, all participants visited with the allocation bottles and returned them to the investigators to count the remaining tablets. Participants who took at least 80 percent of calcium, vitamin D2, and the study tablets of all prescriptions were defined as having good compliance. The serum AST and ALT were monitored. All adverse effects were recorded by either interviewing by investigators or self-reporting by the participants. 

### 2.5. Statistical Analysis

Statistical analysis was performed by STATA Version 15.0 (College Station, TX, USA). The results were analyzed by the intention-to-treat statistical method. The baseline characteristics and the laboratory data were analyzed by the Shapiro–Wilk test for normal or non-normal distribution. All quantitative variables were tested for normal distributions. The student *t*-test was used for the comparison of the continuous variables in parametric data. The Mann–Whitney U test was used for the comparison of continuous variables in nonparametric data. In all instances, two-tailed tests of statistical significance were used. For comparison of the bone turnover marker between points in time within the group, a paired *t*-test was used. Data were presented as mean ± standard deviation (SD) and median (minimum and maximum) if the data had a non-normal distribution. The intention to treat analysis was done, and the level of statistical significance was set to *p* < 0.05.

## 3. Results

### 3.1. Study Participants

A total of 52 participants were recruited. All the participants completed the study in the 12th week. All of the patients in the placebo group had good compliance with the assigned tablet, including the calcium and vitamin D2 supplement, based on the counting of the remaining tablets at 12 weeks. Only one in the vitamin E group had poor compliance (compliance rate 30 percent) with no difference in outcome versus the placebo group. A protocol flow chart is shown in Figure 1. 

Table 1 shows the baseline characteristics of the participants in both groups. There were no significant differences in baseline characteristics.

### 3.2. Bone Turnover Marker 

A comparison of the bone turnover markers, CTX and PINP, showed no significant difference between the vitamin E and placebo group at baseline and after 12 weeks of supplementation. However, the mean difference in the bone resorption marker, CTX, at baseline versus 12 weeks were significantly different between the vitamin E group and the placebo group (−0.003 ± 0.09 and 0.121 ± 0.15, respectively (*p* < 0.001)), and no difference was shown in the mean difference in the formative bone marker, PINP between the vitamin E group and placebo group (−0.25 ± 14.02 and 2.4 ± 10.88, respectively (*p*
*=* 0.430)) as shown in Table 2. In addition, the placebo group was showed a significant increase in bone resorption marker, CTX, during the time period, up by 35.3% at 12 weeks compared with baseline (*p* < 0.001). Meanwhile, the vitamin E group showed no significant change of bone resorption marker at 12 weeks compared with baseline (*p* < 0.898), as shown in Figure 2. The bone formation markers in both groups were not significantly changed. 

### 3.3. Adverse Events 

One participant in the vitamin E group had postmenopausal bleeding after 10 weeks of supplementation and was spontaneously relieved. At the 12-week follow-up visit, this adverse event was self-reported by the participants. She had not had any abnormal bleeding in other sites. A pelvic examination was performed, and the results were unremarkable. Transrectal sonography showed 5 mm of endometrial thickness. Her Pap smear was negative for malignancy. Her laboratory investigation showed no abnormalities, including platelet count, hemoglobin level, coagulogram, and renal and kidney function. Fractional curettage was done, and atrophic endometrium was reported. Neither AST nor ALT level was affected after 12 weeks of study in any participants.

## 4. Discussion

A vitamin E, or mixed tocopherol, supplement for twelve weeks can slow down bone resorption in osteopenic postmenopausal women. This is in line with the animal model study, which showed that vitamin E could suppress bone resorption activity and decrease the rate of bone loss [34,35]. Norazlina et al. studied the effect of vitamin E supplement or alpha-tocopherol on the bone of ovariectomized female rats. They revealed that vitamin E and alpha-tocopherol could suppress bone resorption markers compared to placebo [27]. In addition, Johnson et al. have found that vitamin E supplements can decrease the osteoclast number of femoral bone marrow in ovariectomized rats compared to placebo [32]. This supports the hypothesis that vitamin E (both the tocopherols and tocotrienols) can slow down osteoclast activity exacerbated by estrogen deficiency. This hypothesis is also applicable to humans. Michaëlsson et al. [39] have reported a cohort study of the relationship between intake and serum concentrations of alpha-tocopherol and fractures in 61,433 older Swedish women. The results showed that low intake and low serum concentration of α-tocopherol was associated with an increased fracture rate. Moreover, Holvik et al. investigated the association between serum α-tocopherol concentrations and risk of hip fracture during up to 11 years of follow-up based on the Norwegian Epidemiologic Osteoporosis Studies (NOREPOS) cohort study, which included 21,774 men and women aged between 65 and 79 years. They reported an association between low serum concentrations of vitamin E and increased risk of hip fracture [29]. 

As mentioned earlier, there is an association between low serum vitamin E or tocopherol concentration and the risk of osteoporotic fracture in the elderly. To the best of our knowledge, this is the first randomized controlled trial to evaluate mixed-tocopherol supplement effects on the bone turnover marker in postmenopausal women with osteopenia. In addition, we have found that vitamin E or mixed-tocopherol has some osteoprotective effect against the physiologic bone loss across menopausal women. The study confirmed that bone resorption maker, or serum CTX, was maintained in the intervention group, whereas in the placebo group, this marker was increased. Moreover, there is a significant difference between the intervention and the control group in terms of the mean difference. The serum CTX has a trend of decrease in the treatment group: decreased bone resorption marker. In contrast, a significant increase in serum CTX was found in the placebo group. This evidence the anti-resorptive effect of vitamin E on the bone remodeling process. However, an increase in serum CTX in the placebo group at 12 weeks was within the Asian variation in CTX changes, which have reported around 37–42% [40]. Rathnayake et al. reported a systemic review of the bone marker turnover in Asian pre-and postmenopausal women, including 23 studies from China, Japan, India, Korea, Pakistan, and Thailand. The results of these studies showed that the mean serum CTX varied from 0.25 to 0.433 ng/mL (42%) and, within China, 0.25–0.395 ng/mL (37%). While Western study, Cavalier et al. conducted the cohort study of European Biological Variation Study; EuBIVAS, which included six European laboratories from Italy, Norway, Spain, the Netherlands, and Turkey. The results showed the variation of serum CTX among Caucasian women, aged more than 50 years varied from 13.4–45.0 ng/mL (21%) [41]. For this evidence, the difference in genetic, ethnics, lifestyle, and dietary culture of dairy products, that affect bone remodeling could be the reasons for the variation of bone markers between Asian and Caucasians. 

Tocopherol is a potent antioxidant that is extensively used for wellness supplementation. It can decrease ROS and pro-inflammatory processes. Estrogen deficiency in postmenopausal women increases oxidative stress. The raising of ROS induces osteoclastogenesis and stimulates osteoclast activity that causes bone loss [6]. Tocopherol supplement can lower ROS, thus suppressing osteoclastogenesis, osteoclast activity, and osteocyte apoptosis [9,23]. Tocopherol supplementation can also decrease pro-inflammatory cytokines, especially IL-1, IL-6, and TNF-α, which are essential factors of bone resorption in vivo and in vitro. These cytokines stimulate the formation and activity of osteoclasts, leading to excessive bone resorption [3,28,42]. Tocopherol may decrease bone resorption via these two processes.

For the BTMs measurement, we used CTX and PINP, the reference BTMs, recommended by the International osteoporosis foundation, and the International federation of clinical chemistry and laboratory medicine’s bone marker standard working group as the standard bone formation and bone resorption marker in clinical use. In addition, our measurement method was the automated electrochemiluminescence, which has the least analytical variability compared to other methods, such as enzyme-linked immunosorbent assay and radioimmunoassay. However, the treatment with an anti-resorptive agent resulted in an early decrease of bone resorption marker at around 3 months, followed by a reduction of bone formation marker in approximately 6 months [37,43]. Results in line with these have been achieved using several anti-resorptive agents. Those results showed a reduction in serum CTX levels at 3 months of administration include alendronate, ibandronate, risedronate, zoledronic acid, raloxifene, hormone replacement therapy, and denosumab. Moreover, a reduction in bone turnover marker has been demonstrated as an independent predictor of fracture risk reduction [37]. Hochberg et al. reported a meta-analysis of 16 randomized trials of anti-resorptive therapy and the extent of change in BMD and decreased bone turnover markers in postmenopausal women with osteoporosis. They reported an association between reduced risk of non-vertebral fractures, increased BMD, reduced bone resorption marker, or serum CTX. The meta-analysis revealed that the estimated 70% reduction in a bone resorption marker corresponds to a 40% reduction in non-vertebral fracture risk [44]. A daily 400 IU tocopherol supplement seems to have some benefits in altering bone resorption markers compared to placebo, which could benefit fracture risk reduction. 

There was no transaminitis in all participants after 12 weeks of supplementation. However, one participant in the vitamin E group had vaginal bleeding at 10 weeks of intervention. The cause of the bleeding was atrophic endometrium, according to pathologic results. In vivo study found that vitamin E has an antiplatelet and anticoagulating function [45,46,47]. Previous studies found an increased risk of bleeding from vitamin E supplements in patients who take anticoagulants and have vitamin K deficiency. Hence, vitamin E supplementation should be avoided in high-risk groups, given that no such complications were found in a healthy population, even in high dose vitamin E supplement at a dose 2000 IU per day [48,49,50]. 

Managing osteopenia in postmenopausal women to prevent osteoporosis and fracture includes: calcium and vitamin D supplements, appropriate protein intake, weight-bearing exercise, and avoiding risk factors, such as smoking alcohol, caffeine, and falls [18,51]. This study shows that adding vitamin E supplements combined with calcium plus vitamin D can decrease bone resorption, benefiting postmenopausal osteopenic women.

The strength of this study is its design being a randomized, double-blind, placebo-controlled clinical trial that uses intention-to-treat analysis. Most participants were of good nutritional status, as indicated by the average BMI of 22 kg/m^2^ and had high compliance with medication intake and completed follow-up in both groups. Adverse effects were evaluated, including signs, symptoms, and liver function tests.

The follow-up duration was not long enough to detect a significant change of bone formation marker (P1NP). In addition, the baseline vitamin E level of the participants, daily dietary vitamin E intake, and lifestyle, such as exercise, were not evaluated.

We suggest a longer duration of follow-up and different doses of mixed-tocopherol for supplementation in a future study. Bone density or fracture events would be informative to evaluate the effect of vitamin E on osteoporotic fracture prevention. 

## 5. Conclusions

In conclusion, vitamin E (mixed-tocopherol) supplementation in postmenopausal osteopenic women slows down the increase of bone resorption marker (CTX) that may represent bone loss prevention through anti-resorptive activity. No significant adverse effect was found after 12 weeks of supplementation.

## Figures and Tables

**Figure 1 nutrients-13-04226-f001:**
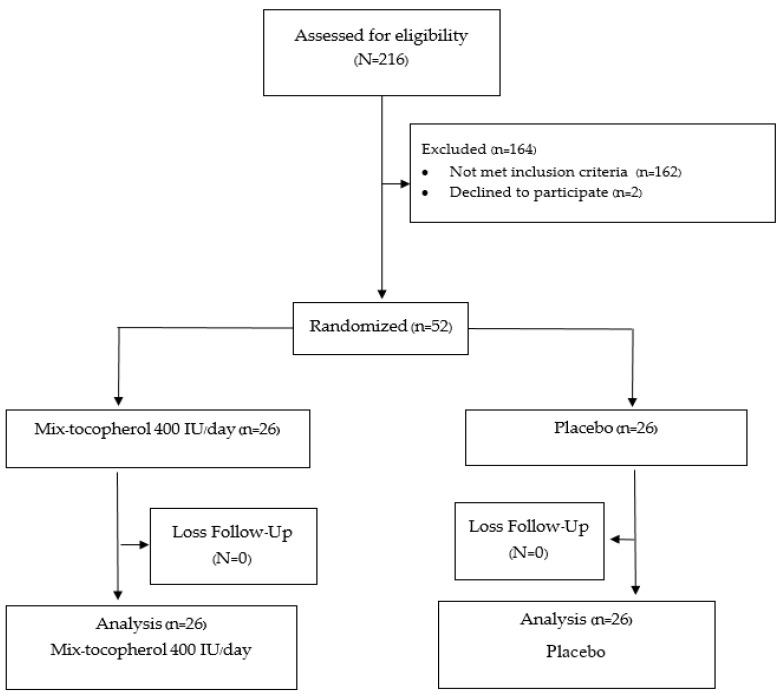
Protocol flow diagram according to CONSORT flow diagram 2010.

**Figure 2 nutrients-13-04226-f002:**
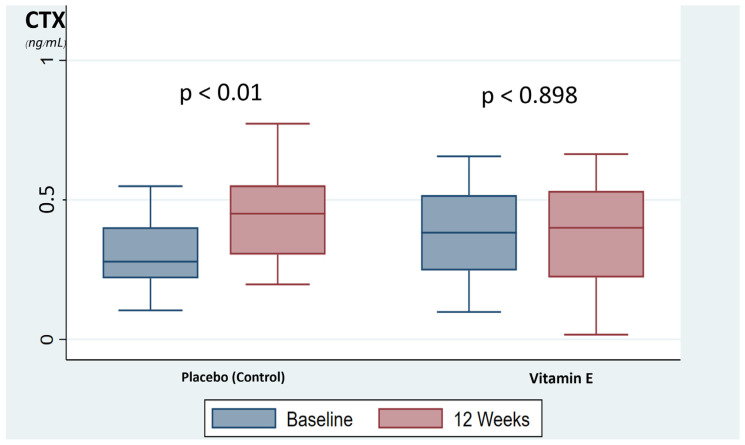
The box-plot represents the changed of bone resorption marker. CTX, C-terminal telopeptide of type I collagen.

**Table 1 nutrients-13-04226-t001:** Baseline characteristics of the participants.

Characteristics	Vitamin E (*n* = 26)	Placebo (*n* = 26)	*p*-Value
Age (years) ^a^	63.15 ± 7.96	62.31 ± 6.50	0.65
Body mass index (kg/m^2^) ^a^	21.93 ± 3.61	22.56 ± 2.73	0.48
Age at menopause (years) ^a^	48.77 ± 4.12	48.58 ± 4.16	0.87
Duration of postmenopausal period (years) ^a^	13.54 ± 7.46	14.58 ± 7.80	0.63
Exercise ^b^			0.58
No exercise	13 (50%)	15 (57.7%)	
Exercise (<150 min/week)	13 (50%)	11 (42.3%)	
Baseline BMD ^a^			
Lumbar spine (g/cm^2^)	0.86 ± 0.12	0.82 ± 0.17	0.15
Femoral neck (g/cm^2^)	0.659 ± 0.07	0.64 ± 0.08	0.63
Total hip (g/cm^2^)	0.75 ± 0.12	0.81 ± 0.08	0.06
AST (U/L) ^a^	23.23 ± 5.96	23 ± 3.17	0.86
ALT (U/L) ^a^	23.11 ± 5.75	22.38 ± 7.46	0.69
Cr (mg/dL) ^a^	0.71 ± 0.15	0.67 ± 0.14	0.24
TSH (mIU/L) ^a^	1.72 ± 0.921	1.44 ± 0.81	0.24
PTH (pg/mL) ^a^	44.5 ± 11.6	51.9 ± 16.57	0.07
25(OH)D (ng/mL) ^a^	34.48 ± 8.08	35.18 ± 10.83	0.79

Notes: ^a^ Data expressed as mean ± standard deviation (SD), ^b^ Data expressed as a percentage. A statistically significant difference in the groups was *p*-value < 0.05. BMD, bone mineral density; AST, aspartate transaminase; ALT, alanine transaminase; Cr, creatinine; TSH, Thyroid-stimulating hormone; PTH, Parathyroid hormone; 25(OH)D, 25-hydroxyvitamin D.

**Table 2 nutrients-13-04226-t002:** Comparison of the bone turnover marker between the vitamin E group and placebo group at baseline and 12 weeks.

Bone Turnover Marker	Vitamin E (*n* = 26)	Placebo (*n* = 26)	*p*-Value
CTX (ng/mL)			
Baseline ^a^	0.39 (0.10, 0.66)	0.28 (0.10, 1.02)	0.17
12 weeks ^a^	0.40 (0.02, 0.66)	0.45 (0.19, 1.16)	0.78
Mean difference ^b^	−0.003 ± 0.09	0.121 ± 0.15	<0.001 *
P1NP (ng/mL)			
Baseline ^a^	50.80 (6.96, 78.90)	52.02 (20.52, 127.9)	0.78
12 weeks ^a^	50.66 (20.43, 78.90)	49.85 (21.17, 139.4)	0.78
Mean difference ^a^	−2.75 (−28.82, 48.41)	4.87 (−27.09, 19.93)	0.10

Notes: ^a^ Data expressed as median(range), ^b^ Data expressed as mean ± standard deviation (SD). CTX, C-terminal telopeptide of type I collagen; P1NP, N-terminal propeptide of type I procollagen. * *p*-value < 0.05 assigned as statistically significant.

## Data Availability

Data available on request due to restrictions on privacy or ethics. The data presented in this study are available on request from the corresponding author. The data are not publicly available due to the ethics and right of the faculty of medicine, Ramathibodi Hospital, Mahidol University.

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
