# Peer review of "Effect of Vitamin E Supplement on Bone Turnover Markers in Postmenopausal Osteopenic Women: A Double-Blind, Randomized, Placebo-Controlled Trial"

_nutrients, 2021, doi:10.3390/nu13124226_

Round 1

Reviewer 1 Report

The search for new therapeutic approaches to prevent bone loss, particularly in women, is an interesting area for research. The submitted manuscript however, has several limitations mainly with regard to the interpretation of results of bone markers reported for the placebo group that requires profound explanation. Without this explanation the manuscript is of little clinical value

Major comments:

1.Bone turnover markers do not present Gaussian distribution therefore the data for BTM should be presented as medians and interquartile ranges (Q1, Q3).  The authors write that non-parametric data are presented as medians but such data are not reported in the article, all presented values are means +/- SD.

2.The authors write they measured bone markers with chemoluminescence , in fact the technology used in CobasE602 is the electrochemiluminescence technology. Moreover, reported CV of the assays for bone markers are surprisingly low.

3.In Table 1 no information on statistical significance, no p values are given which would be of interest. In Table 3 the SD values for means (should be medians) of CTX values are very high, particularly in the placebo group, between 46-62%. This is surprising and not explained by the authors.

  1. The dramatic increase of CTX in the placebo group during 12ve weeks is incredibly high and exceeds significantly, more than two-fold, the biological variation in the general population (approx 16% for CTX); together with CV of the assay this would result in approx 23% of difference  and the reported difference within 12ve weeks is over 35%.  In the EuBIVAS report, 91 subjects were recruited and fasting blood samples were obtained weekly for ten consecutive weeks. The results were subjected to outlier and variance homogeneity analysis before CV-ANOVA was used to obtain the Biological Variation  estimates. The following CVI estimates with 95% confidence intervals (95% CI) were obtained: β-CTX 15.1% (14.4-16.0%) and PINP 8.8% (8.4-9.3%).

Even with the use of efficient antiresorptive therapy the change of CTX within 3 months should be at least 28% and then the continuation of therapy is recommended,  The data from TRIO study reported that among  postmenopausal women only 20% present CTX values higher than the reference values for premenopausal women. The submitted study included postmenopausal women with osteopenia in which bone turnover may be increased , however the baseline CTX in both groups was only slightly higher than the reference for premenopausal ones whereas baseline P1NP was well within the postmenopausal range. It is well known that bone turnover increases in women after menopause and the higher increase is observed between 7-9 years sfter menopause, however the subjects included in the study were in average 14 years after menopause and in this period bone turnover was observed to decline.

The authors definitely have to explain the reason for such high increase (35.3%) of CTX within 12ve weeks in the placebo group. Neither change in BMD nor in 25(OH)D levels may be the explanation. This is even more surprising that no changes of P1NP concentrations within 12ve weeks were observed in both groups taking into consideration that both study groups were supplemented with vitamin  D and calcium. Bone turnover markers P1NP and CTX correlate significantly in adults and  even in children when bone turnover is very dynamic. Did the authors checked correlations between bone markers in their study ?

Final conclusion of the study is based on the fact that dramatic increase of bone resorption was  observed within three months in the placebo group. Without  explanation of the reason for this observation the findings are of little value as definitely there was no  effect of vitamin E supplementation in the treatment group. 

Minor comments:

The section Laboratory measurement should contain only information on the parameters which were measured.

The text should be revised by a native English speaker.

Author Response

November 16, 2021

Dear Academics Editors, Nutrients

Thank you a lot for your valuable comments. I would like to declare my point of view and application from our data based on points your comments.

Question from reviewer : 1.Bone turnover markers do not present Gaussian distribution therefore the data for BTM should be presented as medians and interquartile ranges (Q1, Q3).  The authors write that non-parametric data are presented as medians but such data are not reported in the article, all presented values are means +/- SD.

Answer:  Form reviewers advise. The authors agree and revise to formal statistical reports instead of clinician viewpoints as the reviewer recommends from mean+SD to median and ranges. See table 2 revised except for the mean difference of CTX. The author asks to allow the use of mean+SD due to the mean difference of CTX presenting a strong normal distribution.

Question from reviewer : 2.The authors write they measured bone markers with chemoluminescence , in fact the technology used in CobasE602 is the electrochemiluminescence technology. Moreover, reported CV of the assays for bone markers are surprisingly low.

Answer: Thanks, reviewers, for this detail, It's our minor mistake for referencing the wrong name methods. And for intra- and inter-assay, coefficients of variation are actually standard shown on manuscript page 5, , as a information in the reference picture below.

Question from reviewer : 3.In Table 1 no information on statistical significance, no p values are given which would be of interest. In Table 3 the SD values for means (should be medians) of CTX values are very high, particularly in the placebo group, between 46-62%. This is surprising and not explained by the authors.

Answer: Based on reviewer recommend. The authors revised, corrected and inserted  all of the P-values into table 1 on manuscript page 7.

In Table 3 the SD values for means (should be medians) of CTX values are very high, particularly in the placebo group, between 46-62%. This is surprising and not explained by the authors

Question from reviewer : 4. The dramatic increase of CTX in the placebo group during 12ve weeks is incredibly high and exceeds significantly, more than two-fold, the biological variation in the general population (approx 16% for CTX); together with CV of the assay this would result in approx 23% of difference  and the reported difference within 12ve weeks is over 35%.  In the EuBIVAS report, 91 subjects were recruited and fasting blood samples were obtained weekly for ten consecutive weeks. The results were subjected to outlier and variance homogeneity analysis before CV-ANOVA was used to obtain the Biological Variation  estimates. The following CVI estimates with 95% confidence intervals (95% CI) were obtained: β-CTX 15.1% (14.4-16.0%) and PINP 8.8% (8.4-9.3%).

Answers:

From the data in Asia the variation in CTX changes are around 37-42%, relevant to our results. The reasons are the differences in genetic, ethnics, lifestyle and dietary culture (low calcium diet, low milk intake and cheese), that effects on bone remodeling. For supporting this different results from western countries, the systematic review study in Asia which included 23 studies from China, Japan, India, Korea, Pakistan and Thailand among pre- and post-menopausal women. The mean serum CTX varied from 0.25 to 0.433 ng/mL (42%) and within China 0.25 to 0.395 ng/mL (37%). (Rathnayake H, Lekamwasam S, Wickramatilake C, Lenora J. Variation of urinary and serum bone turnover marker reference values among pre and postmenopausal women in Asia: a systematic review. Arch Osteoporos. 2020 Apr 17;15(1):57.)

Question from reviewer : The authors definitely have to explain the reason for such high increase (35.3%) of CTX within 12ve weeks in the placebo group. Neither change in BMD nor in 25(OH)D levels may be the explanation. This is even more surprising that no changes of P1NP concentrations within 12ve weeks were observed in both groups taking into consideration that both study groups were supplemented with vitamin  D and calcium. Bone turnover markers P1NP and CTX correlate significantly in adults and  even in children when bone turnover is very dynamic. Did the authors checked correlations between bone markers in their study?

Answer:           

From the data in Asia the variation in CTX changes are around 37-42%, relevant to our results as above answer (The reasons are the differences in genetic, ethnics, lifestyle and dietary culture (low calcium diet, low milk intake and cheese), that effects on bone remodeling.)

  • In fact, the change in BMD need at least 1-2 year for significantly change and detected.
  • While the serum 25-hydroxyvitamin D (25(OH)D) levels should not difference between group, due to we need to control this confounding this factor, which also effect on bone merker, before including to this study (the baseline data of 25(OH)D in table 1 not significantly difference between groups.
  • For the PINP need 24 weeks for significantly change, along with the process of bone remodeling, that not shown in our study. As we discussed our limitation on manuscript page 11.
  • Thank you for your carefully mentions about correlation between bone marker. For this point, the authors already checked the correlation between CTX and PINP were analyzed and the results showed positive correlation at 0 week and 12 week correlation were 0.50 (p =0.0002, r225) and 0.65 (p<0.0001, r2 0.47), respectively. Those showed well correlation between CTX and PINP, as reviewer expected.

Final conclusion of the study is based on the fact that dramatic increase of bone resorption was observed within three months in the placebo group. without  explanation of the reason for this observation the findings are of little value as definitely there was no  effect of vitamin E supplementation in the treatment group. 

Answer: Because of our RCT design aimed to proved the hypothesis of vitamin E effect on bone marker in postmenopausal women by controlled and avoided any biases and confounding factors to show only the vitamin E effects on interested bone marker (CTX). The intervention  that we interested was supplement and not supplement vitamin E between study groups, so the results reflected the effect of vitamin E supplement on bone protection within 12 weeks as authors design. For more and longer term effect, bone mineral density and fracture, need furthermore research.

Minor comments:

The section Laboratory measurement should contain only information on the parameters which were measured.

Answer: Thank you for your vauable comments, for this point authors allowed to describe some information about other bone markers for more understanding to readers that not familiar with the research field, and finally detailed on CTX and PINP.

The text should be revised by a native English speaker.

Answer : On your recommend the Native English person, Matthew Brewster Mawson., M.Ed., Owner of Master’s English Laboratory, Bangkok and Marcia Brewster, Senior Consultant at Nautilus International Development Consulting, Inc. and former editor-in-chief, Natural Resources Forum already reviewed our manuscript.

Yours sincerely,

Associate Professor Sakda Arj-Ong Vallibhakara, MD., PhD.(Clinical Epidemiology and Biostatistics), MSIT, MA.IS(Informatics), Board of Preventive Medicine(Epidemiology)

Coresponding;

Orawin Vallibhakara, MD. Corresponding Author,

Head of Menopause Unit, Reproductive Endocrinology and Infertility Unit, Department of Obstetrics and Gynecology, Faculty of Medicine, Ramathibodi Hospital, Mahidol University. Email: [email protected], [email protected]

Phone: +66 86 522 0114

Reviewer 2 Report

The article "Effect of Vitamin E supplement on Bone Turnover Markers in 2 Postmenopausal Osteopenic Women: A double-blind, random-3 ized, placebo-controlled trial" reports the clinical study evaluating the influence of vitamine E suuplementation on bone matabolism in postmenopausal ostoepenic femals. The study was well designed however it was not explained why over 200 patients were checked for their eglibility for the study and only 52 entered the study. The value of the study woud be grater if there were more participants. However the results are interesting and may be of a great value, however they need verification on bigger population. I suggest to discuss the limitations of the study.

Author Response

Thank you for valuable review 

Round 2

Reviewer 1 Report

The authors responded clearly to reviewers' questions but unfortunately did not add in the discussion the explanation as below "From the data in Asia the variation in CTX changes are around 37-42%, relevant to our results. The reasons are the differences in genetic, ethnics, lifestyle and dietary culture (low calcium diet, low milk intake and cheese), that effects on bone remodeling. For supporting this different results from western countries, the systematic review study in Asia which included 23 studies from China, Japan, India, Korea, Pakistan and Thailand among pre- and post-menopausal women. The mean serum CTX varied from 0.25 to 0.433 ng/mL (42%) and within China 0.25 to 0.395 ng/mL (37%). (Rathnayake H, Lekamwasam S, Wickramatilake C, Lenora J. Variation of urinary and serum bone turnover marker reference values among pre and postmenopausal women in Asia: a systematic review. Arch Osteoporos. 2020 Apr 17;15(1):57.)"

It would be very informative for the readers to know precisely the ethnical differences and it would strengthten the Conclusions.

Author Response

November 19, 2021

Dear Academics Reviewer, Nutrients

Thank you for your final feedback.

The authors already included the important information from reviewer suggestion about the variation of bone markers among ethnics between Asian and Caucasian, as showed in the part of discussion; page 9-10 line 304-315. This would be informative and explained the study result of high increase of CTX at 12 weeks in placebo group.

Yours sincerely,

Associate Professor Sakda Arj-Ong Vallibhakara, MD., PhD.(Clinical Epidemiology and Biostatistics), MSIT, MA.IS(Informatics), Board of Preventive Medicine(Epidemiology)

Coresponding;

Orawin Vallibhakara, MD. Corresponding Author,

Head of Menopause Unit, Reproductive Endocrinology and Infertility Unit, Department of Obstetrics and Gynecology, Faculty of Medicine, Ramathibodi Hospital, Mahidol University. Email: [email protected], [email protected]

Phone: +66 86 522 0114

This manuscript is a resubmission of an earlier submission. The following is a list of the peer review reports and author responses from that submission.

Round 1

Reviewer 1 Report

The manuscript submitted to Nutrients entitled “Effect of Vitamin E supplement on Bone Turnover Markers in Postmenopausal Osteopenic Women: A double-blind, randomized, placebo-controlled trial” is an original article which aim to evaluate the effect of mixed-tocopherol supplements on bone remodeling in postmenopausal osteopenic women.

On my opinion the article is well written, with good English. The content of the manuscript is interesting.

I highlighted some issues:

  • English language: minor spell check required.
  • Structured abstract required.
  • Summary of abbreviations required.
  • Introduction: The section is well prepared. Anyway, I would suggest the following improvement:

Bisphosphonate and denosumab treatments remain the first line medical management for prevention of bone fractures, but particular attention should be paid to the risk of developing osteonecrosis of the jaw (ONJ) and the management of this possible complication (https://doi.org/10.1016/j.oooo.2018.11.012 ; https://doi.org/10.1016/j.jcms.2020.01.014 ; https://doi.org/10.1111/dth.13334).

  • Materials and Methods: Please insert number of ethical approval if Authors have it.
  • Results: This section has been properly prepared.
  • Discussion: The discussion of Authors' results is very interesting. Was the nutrition status of patients evaluated? Are the bone turnover markers used reliable??
  • Conclusions: Further studies are needed to confirm Authors’ hypothesis.

After making the indicated changes to the article, it may be suitable for publication.

Thanks for the opportunity to review this manuscript.

Author Response

June  9th , 2021

Dear Editors and Team,

            I am very appreciate for your constructive and helpful suggestion for the manuscript submitted to Nutrients entitled “Effect of Vitamin E supplement on Bone Turnover Markers in Postmenopausal Osteopenic Women: A double-blind, randomized, placebo-controlled trial”. I response to you as your suggestions point by point as follows.

  • The structured abstract required. As I review the instructions for authors” (https://www.mdpi.com/journal/nutrients/instructions#preparation) those write the the abstract should be a single paragraph and should follow the style of structured abstracts, but without headings.

Abstract: (Background) Vitamin E is a strong anti-oxidative stress agent, that affects the bone remodeling process. (Methods) The study is to evaluate the effect of mixed-tocopherol supplements on bone remodeling in postmenopausal osteopenic women. A double-blinded, placebo-controlled randomized trial study was designed to measure the effect of mixed-tocopherol on the bone turnover marker after 12 weeks of the supplementation. (Results) Fifty-two osteopenic postmenopausal women were enrolled and allocated into two groups. The intervention group, received mixed-tocopherol 400 IU/day, while the control group received placebo tablets. Fifty-two participants completed 12 weeks follow up. An intention-to-treat analysis, vitamin E produced a significant difference in the mean difference of bone resorption marker (serum C-terminal telopeptide of type I collagen; CTX) compared with the placebo group (- 0.003 + 0.09 and 0.121 + 0.15, respectively, p < 0.001). In the placebo group, the CTX had significantly increased up to 35.3% at 12 weeks of supplement from baseline (p < 0.001), while in the vitamin E group, there was no significant change of bone resorption marker (p < 0.55). (Conclusion) In conclusion, vitamin E (mixed-tocopherol) supplementation in postmenopausal osteopenic women may represent the preventive effect on bone loss through anti-resorptive activity

  • Summary of abbreviation required was added on page 10- 11, before the reference.
  • Introduction: the first line medical management for fracture prevention and their limitations were added on page 2-3.
  • Material & methods: The number of ethical approval was inserted on page 3, in part of study design.
  • Discussion: I emphasized that the nutrition status of patients was evaluated and mentioned on page 10 line 333-334. And, the bone turnover marker used was reliable as explained in details on page 4-5, in part of Data collection and measurements. Moreover, the BTMs reliability also emphasis again in discussion part on page 10, line 310-315.

Yours sincerely,

Orawin Vallibhakara, MD.
Corresponding Author,

Head of Menopause Unit, Reproductive Endocrinology and Infertility Unit, Department of Obstetrics and Gynecology, Faculty of Medicine, Ramathibodi Hospital, Mahidol University. Email: orawinra38@gmail.com, Orawin.val@mahidol.ac.th

Phone: +66 86 522 0114

Reviewer 2 Report

In this manuscript, the authors demonstrated the effect of mixed-tocopherol supplements on bone remodeling in post-15 menopausal osteopenic women. They measured only two bone turnover markers, CTX and P1NP. However, the authors need to confirm other biomarkers to evaluate the effect of vitamin E on bone turnover. 
(R Civitelli, et al. Bone turnover markers: understanding their value in clinical trials and clinical practice. Osteoporos Int. 2009;20(6):843-51.) 

Author Response

June  9th , 2021

Dear Editors and Team,

            I am very appreciated for your constructive and helpful suggestion for the manuscript submitted to Nutrients entitled “Effect of Vitamin E supplement on Bone Turnover Markers in Postmenopausal Osteopenic Women: A double-blind, randomized, placebo-controlled trial”. I response to your suggestions for the standardized and reason for choosing the CTX and PINP in this study.

            The rationale of use BTMs as the reliable outcomes were discussed and described in both material and methods, in part of Data collection and measurements on page 4-5 and discussion part on page 10, line 310-315.

Yours sincerely,

Orawin Vallibhakara, MD.
Corresponding Author,

Head of Menopause Unit, Reproductive Endocrinology and Infertility Unit, Department of Obstetrics and Gynecology, Faculty of Medicine, Ramathibodi Hospital, Mahidol University. Email: [email protected], [email protected]

Phone: +66 86 522 0114

Round 2

Reviewer 2 Report

In my opinion, the manuscript has not been strengthened, and not reflect my review comments. This manuscript is not suitable for publication in Nutrients.